# InfoFlow KV: Information-Flow-Aware KV Recomputation for Long Context

**Xin Teng** [1 2 *] **Canyu Zhang** [1 2 *] **Shaoyi Zheng** [1 2 *] **Danyang Zhuo** [3] **Tianyi Zhou** [4] **Shenji Wan** [1 2]

## Abstract

Retrieval-augmented generation (RAG) for long-context question answering is bottlenecked by inference-time prefilling over large retrieved contexts. A common strategy is to precompute key–value (KV) caches for individual documents and selectively recompute a small subset of tokens to restore global causal dependencies, but existing methods rely on heuristics or representation discrepancies without modeling whether selected tokens can effectively influence generation. We cast selective KV recomputation as an information flow problem and show that a simple attention-norm signal from the query reliably identifies tokens that are both semantically relevant and structurally positioned to propagate information, when computed under an inference-consistent RoPE geometry. We therefore reconstruct global positional assignments for retrieved chunks and introduce an information-flow–guided chunk reordering strategy. Experiments on Large Language Model and Vision-Language Model benchmarks demonstrate consistent gains over prior methods under comparable latency.

## 1. Introduction

Retrieval-augmented generation (RAG) (Lewis et al., 2020) has become a dominant paradigm for question answering (QA) with large language models (LLMs) and vision–language models (VLMs). In RAG, the model repeatedly retrieves and conditions on large sets of external documents, images, or multimodal evidence, often spanning tens or even hundreds of thousands of tokens, while producing only short answers. In such settings, inference efficiency is primarily constrained by the prefilling stage that computes key–value

(KV) caches over the retrieved context, rather than by autoregressive decoding. This bottleneck is further exacerbated in interactive or multi-query scenarios, where large portions of retrieved context may be reused but naïve full-context prefilling still recomputes the same evidence for each query.

This paper focuses on the multi-chunk KV reuse regime, rather than the setting where full-context attention is impossible because the input exceeds the model context window. In this regime, a natural efficiency strategy is to precompute KV caches for individual documents or evidence chunks offline and assemble the retrieved context at query time by reusing these cached states. Full-context prefilling remains an accuracy reference, but it introduces redundant computation when the same retrieved evidence is served repeatedly. However, after retrieval, autoregressive decoding requires a single globally ordered sequence with causal dependencies, whereas each document's KV cache is computed independently under a local causal mask.

To mitigate this mismatch, existing methods introduce selective KV recomputation, in which a small subset of tokens is recomputed under the global causal mask to partially restore cross-document interactions while retaining most of the efficiency gains of offline precomputation. Representative approaches differ primarily in how recomputation targets are selected. CacheBlend (Yao et al., 2025) identifies tokens whose representations change the most by measuring output discrepancies between cached and full-context runs in early transformer layers; however, for efficiency reasons, this comparison is limited to only a few shallow layers and may not reflect a token's ultimate influence on generation. In contrast, EPIC (Hu et al., 2024) adopts fixed positional heuristics, selecting tokens at predefined positions (e.g., document boundaries) for recomputation, independent of content or query. As a result, neither approach fully captures token importance from two critical perspectives: (i) semantic relevance, particularly with respect to the query, and (ii) effective information flow, i.e., whether a token is structurally positioned to influence downstream decoding under the global causal attention graph. This limitation motivates a more principled recomputation criterion that explicitly aligns token selection with the information pathways required for answer generation.

In this work, we view selective KV recomputation as an

*Equal contribution [1]New York University [2]New York University Shanghai [3]Duke University [4]Mohamed bin Zayed University of Artificial Intelligence. Correspondence to: Shengjie Wang <sw5973@nyu.edu>.

*Proceedings of the $43^{rd}$ International Conference on Machine Learning*, Seoul, South Korea. PMLR 306, 2026. Copyright 2026 by the author(s).

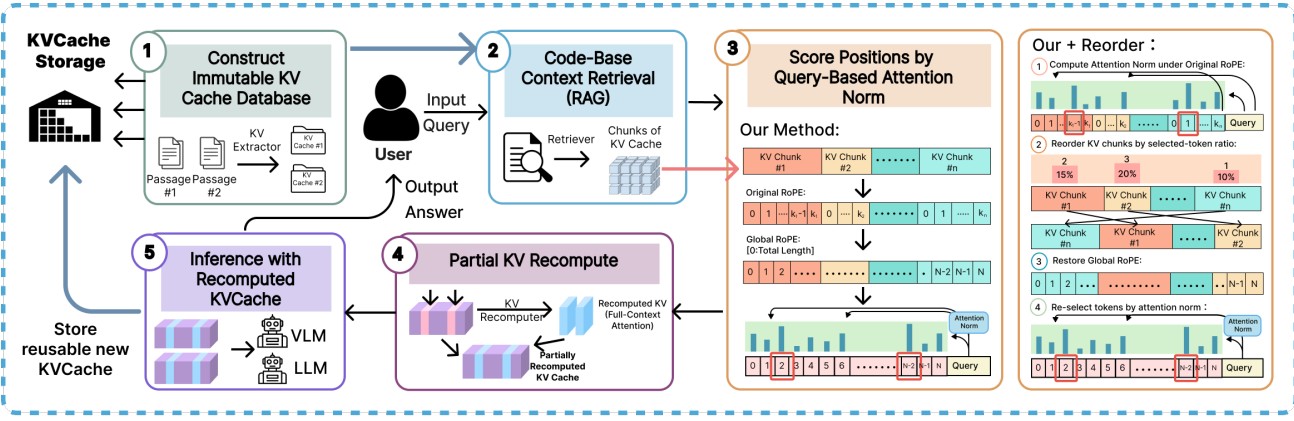

*Figure 1.* Context chunks are prefetched independently using chunk-local RoPE. At inference time, retrieved chunks are concatenated with the prompt, global RoPE positions are reconstructed, and prompt-conditioned attention norms are used to select high-impact tokens for full-context KV recomputation. The recomputed KV states are concatenated with cached chunks, restoring cross-chunk interactions. An optional chunk reordering step places more informative chunks closer to the prompt.

information-flow problem, where the goal is to recover the pathways through which retrieved evidence can effectively influence answer generation. We use "information flow" operationally: it refers to a selected token's ability to affect downstream generation through the model's attention mechanism, rather than to a formal causal analysis of all hidden-state interactions. We find that a simple attention-norm criterion from the query to context tokens is highly effective for identifying recomputation targets. This signal simultaneously captures semantic relevance, by reflecting query–token affinity, and effective information flow, by identifying tokens that are structurally positioned to propagate information under the global causal attention induced by retrieval and decoding. Crucially, this property holds only when attention norms are computed under a positional ordering consistent with inference-time decoding. For RoPE-based models, this requires reconstructing global positional assignments for retrieved chunks; chunk-local or mismatched positional configurations yield unstable and misleading token rankings. Building on this insight, we further propose a retrieved-chunk reordering strategy for independent chunks, improving the effectiveness of downstream attention and providing additional validation of our information-flow perspective. Extensive experiments on both Large Language Model and Vision-Language Model benchmarks show that our approach consistently outperforms existing recomputation methods under comparable budgets.

While our primary focus is retrieval-augmented generation with precomputed document-level KV caches, the proposed chunking-and-recompute strategy also applies when such caches are unavailable. In non-RAG long-context inference, the input can be partitioned into chunks whose KV caches are computed independently, followed by selective recomputation under the global causal mask. This use case is most appropriate when chunk-wise prefilling can be amortized across multiple queries or parallelized across devices; it is not intended to replace exact full-context prefilling when the latter is already sufficiently cheap.

We summarize our contributions as follows.

- We propose a simple and effective *attention-norm-based* criterion from the query to context tokens that jointly captures semantic relevance and effective information flow under the global causal attention graph.

- We show that reliable recomputation target selection requires an *inference-consistent RoPE ordering*, and introduce a global positional reconstruction of retrieved chunks to enable stable and meaningful token ranking.

- Building on the identified information-flow-critical tokens, we further propose a *retrieved-chunk reordering strategy* for independent chunks that improves downstream attention effectiveness, providing additional validation of our information-flow perspective.

- We evaluate our approach on both LLM and VLM RAG benchmarks, demonstrating consistent improvements over existing selective recomputation methods under comparable efficiency budgets.

## 2. Related Work

### 2.1. KV Eviction & Compression

KV eviction and compression methods reduce long-context inference cost by limiting the size or precision of KV caches during decoding. Eviction-based approaches such as H2O (Zhang et al., 2023) and StreamingLLM (Xiao et al.,

2023) discard cached KV states based on heuristics, including recency or attention importance. Compression methods such as Pyramid KV (Cai et al., 2024), SnapKV (Li et al., 2024) and memorization-based approaches (Wu et al., 2022) approximate historical KV representations through hierarchical compression or summary memory. These methods primarily target the memory footprint and access cost during autoregressive decoding. In contrast, our work addresses the prefilling bottleneck under chunk-wise processing, where KV states are computed independently, and cross-chunk interactions are lost. Rather than shrinking or discarding KV caches, we retain token-level KV states and selectively recompute a small subset of tokens to restore global information flow, further influencing downstream generation. Additional comparisons with KV compression and eviction baselines are provided in Appendix D.2.

## 2.2. Chunked Input Processing

Chunked input processing has been explored to extend effective context length beyond the model's native window. Context Expansion with Parallel Encoding (CEPE) (Yen et al., 2024) encodes long contexts in parallel using a lightweight encoder and enables a decoder to attend to chunk representations via cross-attention. A key limitation of such approaches is that chunks are encoded independently, and cross-chunk interactions are not explicitly modeled within the chunk representations. Our method instead targets interaction across chunks by selectively recomputing KV states under the full context. Moreover, CEPE requires additional training to align the parallel encoder with the decoder, whereas our approach operates in a plug-and-play manner at inference time without modifying the pretrained model.

## 2.3. KV Recomputation

KV recomputation methods recover interactions across independently prefetched KV caches by selectively recomputing a small subset of tokens under the full context with a global causal mask. CacheBlend identifies tokens whose cached KV states deviate most from full-context attention, while EPIC mitigates positional mismatch by recomputing fixed chunk-initial tokens. These approaches select recomputation targets through token-level deviation or positional heuristics, without explicitly modeling information flow. Our method is related to prompt-guided token selection in that it uses query-conditioned attention as a relevance signal; however, the central distinction is that we study this signal in the multi-chunk KV reuse setting, where the positional geometry used during scoring must match the geometry used during inference. Unlike methods that treat selection as relevance estimation, our method se- lects tokens based on their capacity to transmit information across chunks, jointly accounting for semantic relevance and positional influence under inference-time attention geometry.

## 2.4. Retrieval Augmented Generation

Retrieval-augmented generation (RAG) (Lewis et al., 2020) improves generation by first retrieving external evidence for a query and then using the retrieved evidence as the model context. Our work is orthogonal to the retrieval component. We do not modify the retriever, the document index, or the ranking objective; instead, we assume that a RAG pipeline has already produced a set of retrieved documents, passages, images, or other evidence. These retrieved items become the chunks used by InfoFlow KV: their KV caches can be prefetched or reused, after which our method selects token-level recomputation targets and, when the chunks are exchangeable, optionally reorders them to improve information flow during generation. Consequently, stronger retrievers can be used directly with our method, while our contribution focuses on efficient and RoPE-consistent inference over the retrieved context. Additional comparisons with retrieval-based baselines are provided in Appendix D.1.

## 3. Preliminary

### 3.1. KV Cache

We consider an autoregressive Transformer decoder. Let $\mathbf{h}_t^\ell \in \mathbb{R}^d$ denote the hidden state at layer $\ell$ and position $t$. For each attention head, keys and values are computed as linear projections of the hidden states,

$$\mathbf{k}_t^\ell = \mathbf{W}_K^\ell \mathbf{h}_t^\ell, \qquad \mathbf{v}_t^\ell = \mathbf{W}_V^\ell \mathbf{h}_t^\ell, \qquad (1)$$

where the head index is omitted for clarity.

During autoregressive decoding, to avoid recomputing keys and values for previously processed tokens, Transformer decoders maintain a key–value (KV) cache that stores all past $\{\mathbf{k}_s^\ell, \mathbf{v}_s^\ell\}_{s \leq t}$ at each layer. We denote the cached tensors up to position $t$ by

$$\mathbf{K}_{\leq t}^\ell \in \mathbb{R}^{t \times d_h}, \qquad \mathbf{V}_{\leq t}^\ell \in \mathbb{R}^{t \times d_h}, \qquad (2)$$

where $d_h$ is the per-head dimension. In practice, inference consists of a *prefilling* phase that computes KV states for an input context of length $L$ in a single forward pass, followed by an iterative *decoding* phase that appends one new KV pair per generated token. In long-context question answering, where the generated answer is short, inference cost is often dominated by the prefilling stage.

### 3.2. Rotary Positional Embedding (RoPE)

Rotary positional embedding (RoPE) encodes positional information by applying position-dependent rotations to queries and keys. For vectors decomposed into 2D subspaces, RoPE applies

$$\text{RoPE}(\mathbf{x}, t) = \begin{bmatrix} \cos(\theta_i t) & -\sin(\theta_i t) \\ \sin(\theta_i t) & \cos(\theta_i t) \end{bmatrix} \mathbf{x}, \qquad (3)$$

where the angular frequencies are defined as

$$\theta_i = 10000^{-2i/d}, \quad i = 0, \dots, \frac{d}{2} - 1. \qquad (4)$$

Lower-index dimensions correspond to higher frequencies and are more sensitive to positional differences, while higher-index dimensions encode lower-frequency components. Consequently, the effectiveness of attention interactions depends on the absolute positions at which tokens are placed, which becomes particularly relevant when KV states are computed independently and later reused or recomputed under different positional assignments.

# 4. Method

In this section, we introduce our method for selecting recomputation targets by jointly considering token semantic relevance and positional influence, with the goal of facilitating information propagation during decoding.

We emphasize that our notion of information flow is an operational one defined over KV-cache reuse and recomputation, rather than a direct measurement of the model's internal semantic information flow. We study whether the KV states of selected context tokens can effectively contribute to downstream decoding after independently prefetched chunks are assembled into a global sequence.

## 4.1. Input Chunking and Prefilling

Let the input consist of $N$ tokens, which we partition into $K$ disjoint chunks $\{C_1, \dots, C_K\}$. Each chunk serves as a basic unit for prefilling and can correspond to a naturally independent segment (e.g., a document or an image), or a contiguous partition of a single long input (e.g., a text sequence or a visual token grid).

During chunk-wise prefilling, each chunk is processed independently to compute its key–value (KV) cache. Within a chunk $C_i$, positional indices are assigned locally: tokens are indexed from 0 to $|C_i| - 1$ according to their order inside the chunk, regardless of their absolute position in the full input. Keys and queries are then computed using RoPE based on these chunk-local positions, and the resulting KV states are stored in the cache associated with $C_i$. As a result, chunk-wise prefilling encodes KV states under chunk-local positional geometry. When chunks are later recombined or recomputed under global positional assignments, this mismatch in positional encoding motivates the need for selective KV recomputation.

## 4.2. Token Selection and Recomputation

**Selection Method** Token selection is performed after the prefetched chunks are concatenated with the prompt, using the KV states obtained from the prefilling stage.

To formalize token selection, we distinguish between chunk-local positions used during prefilling and global positions used during token selection and decoding. Unless optional reordering is applied, the global ordering of chunks is determined by retrieval ranking in RAG settings or by the natural ordering of documents and images. For a context token $t \in C_i$, its global position during token selection is given by

$$g(t) = \Delta_i + p_i(t), \qquad (5)$$

where $p_i(t)$ is the token's chunk-local index and $\Delta_i$ is the starting position of chunk $C_i$ in the concatenated context. Similarly, prompt tokens are assigned global positions

$$g(p) = \Delta_{\mathrm{pr}} + p_{\mathrm{pr}}(p), \qquad (6)$$

where $p_{\mathrm{pr}}(p)$ denotes the prompt-internal ordering and $\Delta_{\mathrm{pr}}$ is the prompt offset.

Under a given positional configuration, recomputation targets are selected based on prompt-conditioned attention norms. Let $\mathbf{A} \in \mathbb{R}^{M \times N}$ denote the prompt-to-context attention matrix at a given layer, where $M$ is the prompt size and $A_{ij}$ represents the attention weight from the $i$-th prompt token to the $j$-th context token. For each context token $j$, we define its importance score as the aggregated attention mass received from all prompt tokens,

$$s_j = \sum_{i=1}^{M} A_{ij}. \qquad (7)$$

Recomputation targets are then selected as the top-$k$ context tokens with the largest importance scores,

$$\mathcal{S} = \text{Top-}k\big(\{s_j\}_{j=1}^{N}\big). \qquad (8)$$

We use prompt-to-token attention rather than token-to-prompt attention as the selection signal, because prompt-to-token attention directly determines how much information is retrieved from each context token during decoding, and therefore has an immediate impact on next-token prediction. This choice is also consistent with the autoregressive nature of language models, where each generated token attends to previously cached keys and values. Intuitively, tokens that receive larger attention mass from the prompt are more likely to contribute to subsequent generation, making them natural candidates for recomputation.

**KV Recomputation** Given the selected set of recomputation targets, we perform KV recomputation under the full global context. Specifically, for each selected token, we recompute its key and value representations by running a standard forward pass with the token placed at its global position in the concatenated sequence. The recomputed KV states then replace their chunk-local cache, while all non-selected tokens reuse their prefetched KV states.

**RoPE Geometry**   Based on the formulation above, token selection under chunk-wise prefilling is influenced by how the prompt and context are positionally encoded after they are concatenated. In particular, the effectiveness of information transmission from context tokens to downstream autoregressive decoding depends on two intrinsic geometric properties of the resulting RoPE assignment: (i) the RoPE frequency range in which context tokens are placed, and (ii) the relative RoPE positional proximity between the prompt and the context. The first property determines how sensitive relative positional differences between tokens are encoded by RoPE, while the second governs whether prompt-to-context attention can be effectively established under causal decoding. These properties are not treated as experimental factors, but as structural aspects of how the selection method is instantiated under different positional assignments.

Within this formulation, we consider four representative RoPE allocation configurations that correspond to distinct positional arrangements under chunk-wise processing. **(1) Global Positioning (GLOBAL).** Both context and prompt tokens are assigned RoPE positions according to their absolute indices in the full global sequence. This configuration reflects the positional geometry encountered in full-context prefilling. **(2) Head-Local Context + Head Prompt (HL–HP).** Context chunks are assigned chunk-local RoPE positions with $\Delta_{\text{ctx}}$ starting from zero, and the prompt is placed immediately after the chunked context. This configuration places both context and prompt tokens in the low-index (high-frequency) RoPE range while maintaining close positional proximity. **(3) Head-Local Context + Tail Prompt (HL–TP).** Context chunks are the same as method (2), while the prompt is placed at its global position index. This configuration increases positional distance between the prompt and the context and places them in different frequency regions. **(4) Tail-Local Context + Tail Prompt (TL–TP).** Prompt tokens are assigned RoPE positions according to their global indices, and all context chunks are placed immediately before the prompt. This configuration pushes context tokens toward higher-index (low-frequency) RoPE regions, while preserving their relative ordering with respect to the prompt. Taken together, these four configurations define a compact yet expressive set of RoPE allocation patterns under chunk-wise processing, covering the principal ways in which frequency placement and prompt–context proximity can arise in practice.

### 4.3. Chunk Reordering

Beyond KV prefilling and token selection, we further consider the ordering of chunks at inference time as an additional degree of freedom for facilitating information flow. Under RoPE-based causal decoding, tokens placed closer to the prompt in the sequence are more likely to interact with prompt queries through attention, which facilitates more

effective propagation of contextual information to downstream generated tokens. We therefore introduce an optional chunk reordering step for settings where input chunks correspond to **independent segments**, such as multi-document retrieval, and where no intrinsic sequential order needs to be preserved. For inputs with intrinsic sequential structure, such as a single long document, reordering may disrupt coherence and is therefore not used.

To enable chunk reordering, we first derive chunk-level importance scores from a first-stage token selection pass. Let $\rho$ denote the recomputation ratio and $k = \lfloor \rho N \rfloor$ the total recomputation budget. We score tokens using the local RoPE geometry induced by independently prefetched chunks and select a single global top-$k$ set $\mathcal{S}^{(1)}$ across all chunks; no per-chunk quota is imposed. For each chunk $C_i$, we define its importance score as

$$r_i = \frac{|\mathcal{S}^{(1)} \cap C_i|}{|C_i|}. \tag{9}$$

Chunks with larger $r_i$ contain a higher density of information-flow-critical tokens. We therefore arrange the context so that higher-scoring chunks are closer to the prompt, while preserving the original order among chunks with equal scores. The benefit of this reordering is further supported by the chunk reordering ablation in Appendix E, where placing informative chunks closer to the query improves prompt–context interaction under causal decoding.

After reordering, we reconstruct GLOBAL RoPE positions according to the new concatenated order and reselect tokens with the same global budget $k$ to obtain the final recomputation set. This second selection pass accounts for the fact that reordering changes the prompt–context RoPE geometry, ensuring that the final selected tokens remain important under the layout used for decoding. Autoregressive decoding is then performed using the reordered chunks and the corresponding recomputed KV states.

Thus, chunk reordering leverages the freedom in chunk ordering to further align RoPE geometry with information flow in the decoding phase, complementing token selection under chunk-wise prefilling. More details on KV recomputation implementation can be found at Appendix F.

## 5. Ablation Results

### 5.1. RoPE Geometry

We ablate different RoPE geometry configurations to study how positional allocation during token selection affects downstream performance.

Table 1 shows that RoPE geometry has a substantial impact on recomputation effectiveness, as the performance shows great differences. Among the tested configurations,

GLOBAL consistently achieves the best performance across benchmarks. In contrast, configurations that assign prompt and context tokens to separate or truncated positional ranges (e.g., HL–HP and TL–TP) result in noticeably lower scores. This suggests that mismatches between the positional layout used for selection and the actual decoding order can degrade the quality of selected recomputation targets.

These results indicate that effective selective recomputation benefits from evaluating token importance under a positional layout that closely matches inference-time decoding. Based on this observation, we adopt GLOBAL as the default RoPE geometry for token selection in our method. In the reordering setting, we use the local-context geometry (HL–TP) for the first-stage selection to avoid favoring chunks solely because of their current global positions, followed by GLOBAL for the second-stage selection to refine recomputation targets under the final positional layout.

*Table 1.* Ablation of RoPE geometry configurations on Qwen under the passage-split setting. Higher is better.

| Method | 2WikiMQA | MuSiQue | HotpotQA | NarrativeQA |
|--------|----------|---------|----------|-------------|
| HL–HP | 0.4455 | 0.2871 | 0.5529 | 0.1481 |
| TL–TP | 0.4458 | 0.2970 | 0.5693 | 0.1923 |
| HL–TP | 0.4722 | 0.3072 | 0.5651 | 0.2106 |
| GLOBAL | **0.5019** | **0.3386** | **0.5954** | **0.2288** |

## 5.2. RoPE Similarity

To further understand why our selection strategy facilitates effective information propagation, we analyze the RoPE similarity between prompt tokens and the selected context tokens. Importantly, this analysis blocks the contribution of token semantics: similarities are computed purely from the RoPE embedding matrices. Specifically, we measure the similarity between the RoPE embeddings of prompt tokens and those of the selected context tokens, and report both the Mean-of-Max (MoM) similarity and the maximum similarity across selected tokens. Higher values indicate that selected tokens occupy RoPE frequency bands that are more closely aligned with the prompt, and are therefore more positionally reachable under causal decoding.

| Model | Method | 2WikiMQA | | HotpotQA | |
|-------|--------|------|------|------|------|
| | | MoM | Max | MoM | Max |
| *LLaMA* | Norm-based | **0.5324** | **0.9773** | **0.5219** | **0.9766** |
| | CacheBlend | 0.5243 | 0.9133 | 0.5191 | 0.8570 |
| | EPIC | 0.5049 | 0.6734 | 0.4985 | 0.6852 |
| *Qwen* | Norm-based | **0.4548** | **0.9805** | 0.4179 | **0.9805** |
| | CacheBlend | 0.4505 | 0.8078 | **0.4226** | 0.8891 |
| | EPIC | 0.4129 | 0.5656 | 0.3835 | 0.6555 |

*Table 2.* RoPE similarity statistics on long-context QA benchmarks. We report Mean-of-Max (MoM) and Max scores across different inference strategies. Higher is better.

As shown in Table 2, norm-based selection generally yields higher RoPE similarity scores than prior methods across both datasets and model families, under both the Mean-of-Max and Max metrics. This result reveals an alignment between attention salience and RoPE-induced positional structure. Specifically, tokens with high prompt-conditioned attention norms tend to occupy positions that are also favorable for information transmission in RoPE space. This observation suggests that norm-based selection naturally identifies recomputation targets that are efficient from a purely geometric perspective, even without explicitly incorporating RoPE similarity into the selection criterion.

## 6. Experiments

### 6.1. Experiment Setup

**Methods.** We compare our approach against a set of representative baselines and prior KV recomputation methods. Across all experiments, all recomputation-based methods use a fixed recomputation ratio of 15% unless otherwise specified. We evaluate the following inference strategies: (i) *Baseline*, which performs full-context prefilling without chunking and serves as the full-computation accuracy reference; (ii) *No Recompute*, which applies chunk-wise prefilling without any KV recomputation; (iii) *Our*, the proposed semantic- and position-aware selective KV recomputation method; (iv) *Our + Reorder*, which further incorporates the segment reordering strategy; (v) *CacheBlend*; and (vi) *EPIC*, which recomputes a fixed proportion of tokens.

**LLM Benchmarks.** We conduct long-context question answering experiments on three large language models: Qwen3-14B (Yang et al., 2025), Llama-3.1-8B-Instruct (Grattafiori et al., 2024), and GLM-4-9B (GLM et al., 2024). Experiments are performed on four datasets from LongBench: 2WikiMQA (Ho et al., 2020), MuSiQue (Trivedi et al., 2022), HotpotQA (Yang et al., 2018), and NarrativeQA (Kočiský et al., 2018). All datasets are evaluated using their official evaluation protocols, and we report the standard metrics specified by each benchmark. We also include the Needle-in-the-Haystack analysis in Appendix A to evaluate the recall capacity in long context.

**VLM Benchmarks.** To evaluate the generality of our method in multimodal settings, we further conduct experiments on the vision–language model Qwen3-VL-8B (Bai et al., 2025). We evaluate on five representative vision–language benchmarks: OCRBench (Liu et al., 2024), ChartQA (Masry et al., 2022), RealWorldQA (xAI, 2024), HRBench4K (Wang et al., 2025), and InfoVQA (Mathew et al., 2022). All results are obtained following the official evaluation protocols of the respective datasets.

*Table 3.* Task performance comparison across different LLMs. Results are reported under a fixed recomputation ratio of 15%, fixed chunk size of 2048 and passage-split settings. Higher is better. Best and second-best results are highlighted excluding baseline and no-recompute methods.

| Model | Method | Fixed Chunk (2048) | | | | Passage Split | | | |
|---|---|---|---|---|---|---|---|---|---|
| | | 2WikiMQA | MuSiQue | HotpotQA | NarrativeQA | 2WikiMQA | MuSiQue | HotpotQA | NarrativeQA |
| *Qwen* | Baseline | 0.5161 | 0.3718 | 0.5922 | 0.1654 | 0.5161 | 0.3718 | 0.5922 | 0.1654 |
| | No Recompute | 0.3948 | 0.1342 | 0.4633 | 0.1137 | 0.1162 | 0.1012 | 0.3059 | 0.2078 |
| | Our (15%) | 0.5089 | 0.3384 | 0.5967 | 0.2110 | 0.5019 | 0.3386 | 0.5954 | 0.2288 |
| | Our + Reorder (15%) | 0.4773 | 0.2872 | 0.5053 | 0.2251 | 0.5058 | 0.3285 | 0.5972 | 0.2310 |
| | CacheBlend(15%) | 0.4417 | 0.2611 | 0.5352 | 0.2170 | 0.4330 | 0.2765 | 0.5738 | 0.2197 |
| | EPIC (15%) | 0.4321 | 0.2368 | 0.5284 | 0.1999 | 0.3697 | 0.2480 | 0.5443 | 0.2291 |
| *LLaMA* | Baseline | 0.4588 | 0.3285 | 0.5410 | 0.1862 | 0.4588 | 0.3285 | 0.5410 | 0.1862 |
| | No Recompute | 0.3969 | 0.2523 | 0.4671 | 0.2639 | 0.3066 | 0.2462 | 0.4253 | 0.2810 |
| | Our (15%) | 0.4635 | 0.3104 | 0.5150 | 0.2891 | 0.4208 | 0.2996 | 0.5123 | 0.3141 |
| | Our + Reorder (15%) | 0.4455 | 0.3044 | 0.5053 | 0.2957 | 0.4417 | 0.2793 | 0.5202 | 0.3243 |
| | CacheBlend (15%) | 0.4131 | 0.2823 | 0.4720 | 0.2685 | 0.3976 | 0.2708 | 0.4872 | 0.3095 |
| | EPIC (15%) | 0.4087 | 0.2638 | 0.4755 | 0.2701 | 0.3885 | 0.2852 | 0.4898 | 0.3102 |
| *ChatGLM* | Baseline | 0.5253 | 0.3946 | 0.6003 | 0.3264 | 0.5253 | 0.3946 | 0.6003 | 0.3264 |
| | No Recompute | 0.4370 | 0.2833 | 0.5024 | 0.2758 | 0.3523 | 0.2474 | 0.4388 | 0.3181 |
| | Our (15%) | 0.5064 | 0.3688 | 0.5739 | 0.3239 | 0.4890 | 0.3758 | 0.5614 | 0.3100 |
| | Our + Reorder (15%) | 0.5176 | 0.3786 | 0.5820 | 0.3140 | 0.4666 | 0.3635 | 0.5567 | 0.3180 |
| | CacheBlend (15%) | 0.4226 | 0.2624 | 0.5177 | 0.2970 | 0.3757 | 0.3188 | 0.5164 | 0.3179 |
| | EPIC (15%) | 0.4401 | 0.2902 | 0.5362 | 0.2962 | 0.4521 | 0.3091 | 0.5481 | 0.3142 |

**Efficiency Measurement.** Single-GPU efficiency measurements are conducted on an NVIDIA A100 GPU unless otherwise specified. For each method, we measure end-to-end inference latency and report the median runtime over 10 independent runs. The sequence-parallel experiments in Section 7 use the multi-GPU setup specified there.

## 6.2. LLM Results

**Analysis on LongBench.** Table 3 reports long-context QA performance on LongBench across three LLM backbones under both fixed-chunk and passage-split settings.

Under both evaluation settings, *Our Method* achieves the strongest and most consistent performance among recomputation-based methods. It attains the best or second-best results across all four benchmarks across Qwen, LLaMA, and ChatGLM, compared to previous methods, with particularly strong gains on multi-hop reasoning tasks such as 2WikiMQA, MuSiQue, and HotpotQA. These tasks require aggregating evidence scattered across distant context segments, making them particularly vulnerable to cross-chunk information loss and thereby highlighting the advantage of our method in cross-chunk information transfer.

We further observe that *Our + Reorder* maintains comparable performance and yields additional gains in several passage-split and narrative-style settings, suggesting that reordering helps when chunks are independent, because it places informative chunks closer to the prompt for better

information propagation.

Overall, these results further validate the effectiveness and robustness of our ap- proach on LLMs and underscore the importance of selec- tively recomputing tokens that facilitate information flow across chunks.

## 6.3. VLM Results

**VLM Results Analysis.** Table 4 summarizes performance on Qwen3-VL-8B across multiple vision–language QA benchmarks under different visual chunking levels. Here, k denotes the number of visual chunks used in the chunk-wise prefilling pipeline, not the top-k token-selection budget used for recomputation; within each k setting, all recomputation methods use the same recomputation budget. Larger k creates more independently prefetched chunks and therefore a stronger cross-chunk mismatch, so absolute performance can decrease as the input is partitioned more aggressively. The relevant comparison is therefore between methods under the same k and identical recomputation settings.

Under both $k = 2$ and $k = 4$ visual chunking settings, *Our* method achieves the strongest or most stable performance across benchmarks. In particular, it consistently outperforms CacheBlend and EPIC on RealWorldQA and ChartQA, and surpasses both on OCRBench at k = 4 (at k = 2 it leads EPIC while remaining on par with CacheBlend, 842 vs. 845). The improvements are most evident on structurally demanding tasks such as ChartQA and OCRBench, where

*Table 4.* Performance comparison on Qwen3-VL-8B across multiple vision–language QA benchmarks under different visual chunking levels $k$. Here, $k$ denotes the number of visual chunks in the chunk-wise prefilling pipeline, while the recomputation budget is fixed across methods within each $k$ setting. The $k = 0$ setting corresponds to standard inference without chunk-wise recomputation. Higher is better. Best and second-best results are highlighted excluding baseline and no-recompute methods.

| Model | Method | RealWorldQA | ChartQA | OCRBench | HRBench4K | infoVQA val |
|---|---|---|---|---|---|---|
| *Qwen3-VL-8B* ($k = 0$) | Baseline (No Recompute) | 0.7059 | 83.08 | 878 | 0.74875 | 83.07 |
| *Qwen3-VL-8B* ($k = 2$) | No Recompute | 0.6745 | 71.32 | 839 | 0.72750 | 71.64 |
| | Our | 0.6810 | 73.48 | 842 | 0.72625 | 73.07 |
| | CacheBlend | 0.6758 | 71.72 | 845 | 0.72375 | 72.00 |
| | EPIC | 0.6745 | 70.92 | 836 | 0.72500 | 71.51 |
| *Qwen3-VL-8B* ($k = 4$) | No Recompute | 0.6588 | 62.00 | 781 | 0.68875 | 57.88 |
| | Our | 0.6667 | 65.68 | 802 | 0.69125 | 62.23 |
| | CacheBlend | 0.6562 | 62.68 | 786 | 0.68625 | 58.56 |
| | EPIC | 0.6549 | 62.48 | 785 | 0.67250 | 57.82 |

successful reasoning requires integrating dispersed visual elements with textual cues over long contexts.

Taken together, these results demonstrate the broad applicability of the proposed recomputation strategy: rather than being limited to language-only models, it consistently delivers substantial improvements in vision–language settings across diverse multimodal reasoning tasks.

## 7. Efficiency Analysis

**Latency with Prepared Context** We first consider the inference setting in which the context KV cache is already prepared. This corresponds to scenarios where long contexts are prefetched offline or reused across multiple queries, so that inference primarily consists of selective KV recomputation and autoregressive decoding. Under this setting, latency is dominated by recomputation rather than full-context prefilling, allowing us to directly examine the speed–accuracy trade-off induced by different recomputation strategies. As shown in Fig. 2, we observe a clear Pareto frontier between TTFT and downstream performance. Within curves, our method consistently achieves higher accuracy at comparable latency, or equivalently lower latency at similar performance levels, indicating that recomputation budget is allocated to tokens that most effectively support downstream generation. A detailed runtime breakdown is provided in Appendix G.

**Latency with On-the-Fly Context Prefilling** We next consider a more general setting in which no context cache is available at inference time, and the TTFT includes chunk-wise prefilling followed by selective KV recomputation and decoding. In this regime, the cost of attention computation during prefilling becomes a dominant factor. Empirically, attention execution on a single device falls short of ideal quadratic scaling in practice, limiting the efficiency gains

from chunk-wise prefilling. As a result, the prefilling overhead cannot be effectively amortized without parallelization.

This motivates the use of multi-GPU sequence parallelism, which distributes attention computation across devices during prefilling. By improving the scaling behavior of attention, sequence parallelism substantially reduces prefilling latency and exposes a more favorable efficiency regime for long-context inference. We therefore evaluate recomputation strategies under multi-GPU sequence-parallel (Shoeybi et al., 2019) execution to characterize end-to-end latency behavior when context must be constructed on the fly.

For long-context settings ($\geq$8K tokens), the overhead introduced by KV recomputation becomes small compared to the cost of full-context attention. As shown in Table 5, our method achieves competitive or better TTFT than ring attention (Liu et al., 2023) from 8K onward, and consistently outperforms it at sequence lengths of 16K and above. In particular, at 16K our method reduces TTFT from 707.8 ms (ring attention) to 427.6 ms, and at 32K from 2350.1 ms to 914.0 ms, corresponding to a $2.57\times$ speedup over ring attention and up to a $3.49\times$ speedup over the single-GPU full-prefill baseline. These results show that selectively recomputing a small set of information-critical tokens is far more efficient than attending to the full context, with the overhead rapidly amortized as sequence length grows.

We note that, under the ring-attention setting, our method avoids all-gathering the full KV cache across GPUs for cross-chunk attention. Instead, it communicates only the small subset of tokens selected for recomputation, and most recomputation remains local when a selected token lies in the same sequence-parallel chunk. As a result, inter-GPU communication is significantly reduced, and the latency advantage becomes more pronounced as context length grows, leading to faster prefilling in long-context regimes.

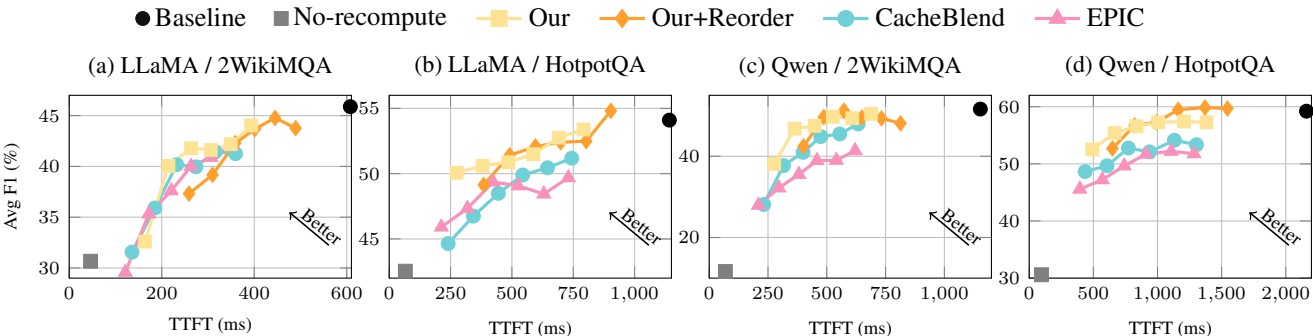

*Figure 2.* Speed–accuracy trade-off on LLaMA and Qwen across long-context QA benchmarks. Each curve corresponds to a recomputation budget sweep. Upper-left indicates a better trade-off.

*Table 5.* TTFT latency and speedup comparison under different sequence lengths with sequence-parallel execution on 4 NVIDIA H100 GPUs. Results for the proposed method are reported under a fixed recomputation ratio of 15%. Speedup is reported relative to single-GPU prefilling.

| Seq Len | Method | TTFT (ms) | Speedup |
|---------|--------|-----------|---------|
| 8192 | Single-GPU Prefill | 566.7 | 1.00x |
| | Ring Attention | 247.5 | 2.29x |
| | Our | **232.0** | **2.44x** |
| 16384 | Single-GPU Prefill | 1285.8 | 1.00x |
| | Ring Attention | 707.8 | 1.82x |
| | Our (15%) | **427.6** | **3.01x** |
| 32768 | Single-GPU Prefill | 3190.5 | 1.00x |
| | Ring Attention | 2350.1 | 1.36x |
| | Our (15%) | **914.0** | **3.49x** |

As shown in Table 6, our method also outperforms ring attention in answer quality under the same sequence-parallel execution setting, improving F1 on all three QA benchmarks. Specifically, we observe absolute F1 gains of +0.0130 on HotpotQA, +0.0255 on 2WikiMQA, and +0.0144 on MuSiQue. These results indicate that selectively recomputing information-critical tokens not only preserves accuracy under sequence parallelism, but can further enhance reasoning performance.

| Method | 2WikiMQA | MuSiQue | HotpotQA |
|--------|----------|---------|----------|
| Ring Attention | 0.4894 | 0.3210 | 0.5653 |
| Our(15%) | **0.5149** | **0.3354** | **0.5783** |

*Table 6.* Comparison between Ring Attention and the proposed method under a fixed recomputation ratio of 15% across long-context QA benchmarks. Under identical sequence-parallel settings, the proposed method achieves consistent F1 improvements.

## 8. Discussion

**Recomputation Efficiency under Irregular Attention Masks** Selective KV recomputation requires attending a dynamically selected subset of tokens to the full context under a causal constraint, resulting in an irregular attention mask that is neither fully dense nor strictly causal. Such patterns are not directly supported by FlashAttention-style dense kernels, so our implementation uses FlashInfer-style kernels designed for sparse or irregular attention, while dense full-prefill and ring-attention baselines use their corresponding optimized kernels. Thus, the reported TTFT comparisons should be interpreted as comparisons between optimized implementations available for the respective attention patterns, not as idealized FLOP counts. In practice, we observe that recomputation can incur up to $2\times$ overhead relative to its ideal compute cost, even at small recomputation ratios. This limitation is shared by selective recomputation methods in general and is primarily a kernel-level issue. Designing customized kernels for sparse, index-based causal attention is a promising direction for future work.

## 9. Conclusion

We revisit selective KV recomputation from an information-flow perspective for chunk-wise long-context inference. By selecting recomputation targets using prompt-conditioned attention norms under inference-consistent RoPE geometry, our method restores effective cross-chunk information transmission without modifying the pretrained model. The resulting method is most useful in multi-chunk KV reuse workflows, where exact full-context prefilling is feasible but unnecessarily redundant. Experiments on both large language models and vision-language models demonstrate consistent improvements under fixed recomputation budgets compared to previous literature, highlighting the importance of information flow-aware token selection and positional alignment in efficient long-context inference.

## Impact Statement

This work studies how to identify which tokens should be recomputed when chunk-wise prefilling is used for long-context inference in large language and vision–language models. Rather than introducing a new efficiency mechanism, our contribution lies in improving the way recomputation targets are selected, by grounding token importance in prompt-conditioned attention and positional structure.

By providing a principled and empirically validated token selection strategy, this work can help make existing recomputation-based inference methods more reliable and predictable across different models, tasks, and input structures. A better understanding of which tokens facilitate information flow may also inform future designs of long-context inference systems and guide more interpretable and controllable approximations.

This work does not introduce new model capabilities, training data, or deployment settings. As a result, it does not create additional risks related to privacy, bias, or misuse beyond those already present in pretrained language and vision–language models. Any downstream impact, therefore, depends on how such models are applied, and existing responsible-use practices remain applicable.

Overall, this work aims to clarify and strengthen the design principles behind token-level recomputation, contributing to more robust and well-understood long-context inference methods.

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

# A. Needle-in-a-Haystack Analysis.

Figure 3 visualizes the robustness of different inference strategies on the Needle-in-a-Haystack task, where a single relevant fact is placed at varying depths within increasingly long contexts. The baseline model maintains near-perfect retrieval accuracy across all depths and context lengths, while chunk-wise prefilling without recomputation exhibits severe degradation as context length increases, indicating a near-complete loss of long-range information access.

Selective KV recomputation substantially alleviates this failure mode. Both Our and Our + Reorder recover high retrieval accuracy across most depths, demonstrating that recomputing a small number of carefully selected tokens is sufficient to restore effective long-range information flow. Compared to CacheBlend and EPIC, our method exhibits fewer failure regions, particularly at larger context lengths, suggesting more stable information propagation under causal decoding. These results further highlight the role of positional allocation and reordering in long-context inference. By implicitly prioritizing tokens that are both prompt-relevant and positionally effective, our method creates a more favorable prompt–context geometry, enabling reliable retrieval even when relevant information is deeply buried within the context.

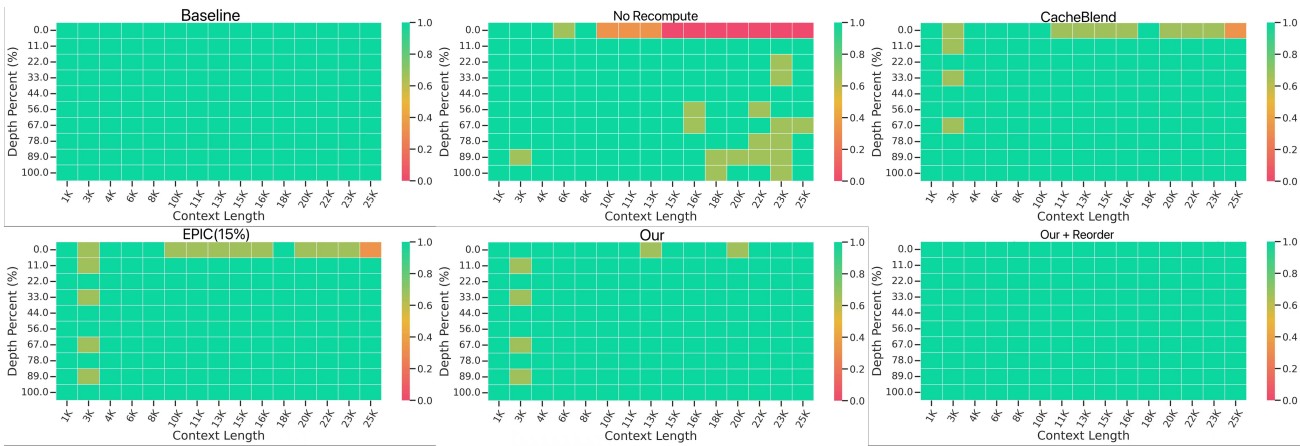

*Figure 3.* Needle-in-a-Haystack accuracy heatmaps on Qwen3-14B under varying context lengths and needle depths.

# B. Additional Baseline Comparisons

## B.1. Comparison with Training-Based Methods

We compare our method with KVLink5 (Yang et al., 2026) using Llama3-3B on four long-context QA benchmarks. As shown in Table 7, our method achieves comparable average performance to KVLink5. However, KVLink5 requires additional training on over 3M datapoints and may be sensitive to data and domain shifts. In contrast, our approach is fully training-free and can be directly applied across models and tasks without retraining.

| Model | Method | 2WikiMQA | MuSiQue | HotpotQA | NarrativeQA | Avg |
|---|---|---|---|---|---|---|
| *Llama-3-3B-Instruct* | Baseline | 0.4073 | 0.3293 | 0.5131 | 0.2762 | 0.3815 |
| | No Recompute | 0.2218 | 0.2161 | 0.3863 | 0.2391 | 0.2658 |
| | Our (15%) | 0.4007 | 0.2693 | 0.4652 | 0.2574 | 0.3482 |
| | Our + Reorder (15%) | 0.3661 | 0.2522 | 0.4994 | 0.2742 | 0.3480 |
| | KVLink5 | 0.4049 | 0.2555 | 0.4551 | 0.2766 | 0.3480 |

*Table 7.* Comparison with KVLink5 on Llama3-3B under a fixed recomputation ratio of 15%. Higher is better. Best and second-best results are highlighted excluding baseline and no-recompute methods.

## B.2. Comparison with Training-Free Baselines

Additional comparisons with TurboRAG (Lu et al., 2025) and PCW (Ratner et al., 2023) are provided on both Llama-3.1-8B-Instruct and Qwen3-14B. As shown in Table 8, the proposed method consistently achieves strong performance relative to prior training-free KV selection and recomputation baselines across multiple long-context QA benchmarks. These results further support the effectiveness of the proposed RoPE-consistent formulation in preserving information flow during generation.

| Model | Method | 2WikiMQA | MuSiQue | HotpotQA | NarrativeQA | Avg |
|---|---|---|---|---|---|---|
| *Llama-3.1-8B-Instruct* | Baseline | 0.4588 | 0.3285 | 0.5410 | 0.1862 | 0.3786 |
| | No Recompute | 0.3066 | 0.2462 | 0.4253 | 0.2810 | 0.3148 |
| | Our (15%) | 0.4208 | 0.2996 | 0.5123 | 0.3141 | 0.3867 |
| | Our + Reorder (15%) | 0.4417 | 0.2793 | 0.5202 | 0.3243 | 0.3914 |
| | TurboRAG(1024) | 0.3640 | 0.2600 | 0.4497 | 0.2190 | 0.3232 |
| | PCW | 0.3073 | 0.2388 | 0.4145 | 0.3292 | 0.3224 |
| *Qwen3-14B* | Baseline | 0.5161 | 0.3718 | 0.5922 | 0.1654 | 0.4114 |
| | No Recompute | 0.1162 | 0.1012 | 0.3059 | 0.2078 | 0.1828 |
| | Our (15%) | 0.5019 | 0.3386 | 0.5954 | 0.2288 | 0.4162 |
| | Our + Reorder (15%) | 0.5058 | 0.3285 | 0.5972 | 0.2310 | 0.4156 |
| | TurboRAG(1024) | 0.2880 | 0.0723 | 0.3337 | 0.0490 | 0.1858 |
| | PCW | 0.1939 | 0.1541 | 0.3806 | 0.2727 | 0.2503 |

*Table 8.* Comparison with training-free baselines under a fixed recomputation ratio of 15%. Higher is better. Best and second-best results are highlighted excluding baseline and no-recompute methods.

# C. Additional Benchmarks

## C.1. LongBench v2

To further evaluate robustness under broader long-context settings, experiments were additionally conducted on LongBench v2 using Qwen3-14B. Evaluation was performed on the short-question subset (10k–210k context length), consisting of 180 samples that can fit within a single-GPU full-prefill setting. This subset was selected because many longer examples require substantial truncation under single-GPU full-prefill inference, where baseline performance often degenerates toward near-random guessing, making comparisons less informative. As shown in Table 9, the proposed method remains competitive under substantially longer contexts, outperforming alternative recomputation and retrieval-based baselines by a clear margin. In particular, the proposed method achieves 28.89% accuracy, compared with 26.11% for CacheBlend and 20.56% for EPIC, suggesting that the proposed information-flow-aware recomputation strategy generalizes effectively across broader long-context benchmarks.

| Model | Method | Accuracy |
|---|---|---|
| *Qwen3-14B* | Baseline | 0.3833 |
| | No Recompute | 0.2111 |
| | Our(15%) | 0.2889 |
| | CacheBlend(15%) | 0.2611 |
| | EPIC(15%) | 0.2056 |

*Table 9.* Results on LongBench v2 short-question subset using Qwen3-14B under a fixed recomputation ratio of 15%. Higher is better. Best and second-best results are highlighted excluding baseline and no-recompute methods.

# D. Additional Clarifications

## D.1. Retrieval-Based Baselines

Our work focuses on post-retrieval KV cache management rather than retrieval itself. Datasets such as HotpotQA and 2WikiMQA already provide retrieved context, and the proposed method performs token-level recomputation within the provided context. Retrieval-based methods are therefore largely orthogonal to the problem setting considered in this work.

To further clarify this distinction, an additional retrieval-based baseline, BM25 Recompute, is included. BM25 Recompute replaces token-level selection with passage-level retrieval under the same recomputation budget. As shown in Table 10, BM25 Recompute consistently underperforms the proposed method across datasets, suggesting that coarse chunk-level retrieval cannot effectively substitute for selective token-level recomputation.

| Model | Method | 2WikiMQA | MuSiQue | HotpotQA | NarrativeQA |
|-------|--------|----------|---------|----------|-------------|
| *Qwen3-14B* | Baseline | 0.5161 | 0.3718 | 0.5922 | 0.1654 |
| | Our | 0.5019 | 0.3386 | 0.5954 | 0.2288 |
| | Our + Reorder | 0.5058 | 0.3285 | 0.5972 | 0.2310 |
| | BM25 Recompute | 0.1904 | 0.1117 | 0.4153 | 0.1874 |

*Table 10.* Comparison with retrieval-based baselines on Qwen3-14B. Higher is better. Best and second-best results are highlighted excluding baseline methods.

## D.2. KV Compression and Eviction Baselines

The proposed method is related to KV compression and eviction approaches such as PyramidKV, SnapKV, and StreamingLLM. However, these methods primarily reduce KV memory through irreversible token removal, while the proposed method selectively recomputes informative KV representations to preserve long-range information flow.

These approaches are complementary rather than mutually exclusive. The proposed method operates during prefilling and can be naturally combined with KV compression or sparse-attention techniques during decoding. As shown in Table 11, the proposed method consistently achieves stronger performance under comparable settings.

| Model | Method | 2WikiMQA | MuSiQue | HotpotQA | NarrativeQA |
|-------|--------|----------|---------|----------|-------------|
| *Llama-3.1-8B-Instruct* | Baseline | 0.4588 | 0.3285 | 0.5410 | 0.1862 |
| | No Recompute | 0.3066 | 0.2462 | 0.4253 | 0.2810 |
| | Our | 0.4208 | 0.2996 | 0.5123 | 0.3141 |
| | Our + Reorder | 0.4417 | 0.2793 | 0.5202 | 0.3243 |
| | PyramidKV | 0.2547 | 0.1457 | 0.2619 | 0.2932 |
| | SnapKV | 0.2643 | 0.1498 | 0.2585 | 0.2911 |
| | StreamingLLM | 0.2423 | 0.1114 | 0.2257 | 0.1932 |

*Table 11.* Comparison with KV compression baselines on Llama-3.1-8B-Instruct. Higher is better. Best and second-best results are highlighted excluding baseline and no-recompute methods.

## D.3. Relation to Full-context Approximation

Full-context approximation refers to the objective of recovering the attention behavior obtained by running standard full-context prefilling over the entire concatenated sequence. This is a natural diagnostic for methods that explicitly aim to approximate full attention under a reduced computation budget. However, this is not the primary objective of our method. Our formulation instead targets effective information flow: selected tokens should be able to influence downstream generation through attention-based interactions under positional geometry that is consistent with inference-time decoding. Therefore, approximation fidelity to full-context attention and information-flow-aware recomputation measure related but distinct properties. Nevertheless, we include this diagnostic to better understand whether our method also recovers

*Table 12.* Attention-pattern recovery relative to full-context attention on HotpotQA. We report the average attention matrix MSE across all layers using 100 samples at a 15% recomputation ratio. Lower is better.

| Model | Method | Attention MSE $(10^{-7})\downarrow$ | Reduction $\uparrow$ |
|-------|--------|-----------------------------------|---------------------|
| *Qwen3-14B* | No Recompute | 4.03 | – |
| | Our | 0.88 | 78.3% |
| | CacheBlend | 1.23 | 69.5% |
| | EPIC | 2.05 | 49.1% |

full-context attention structure. We compute the mean squared error (MSE) between the attention matrices produced by each recomputation method and the corresponding full-context attention matrices, averaged across all layers, using 100 HotpotQA samples at a 15% recomputation ratio. As shown in Table 12, our method achieves the lowest attention MSE, reducing the error from $4.03 \times 10^{-7}$ for No Recompute to $0.88 \times 10^{-7}$, corresponding to a 78.3% reduction. It also improves over CacheBlend and EPIC. These results suggest that, although our method is not designed as a direct full-context attention approximation method, enforcing inference-consistent information flow also helps recover the attention structure of full-context inference.

# E. Chunk Reordering Ablation

To further analyze the effect of chunk placement, two reordering strategies are compared on Qwen3-14B: *Ascending*, which places high-importance chunks closer to the query, and *Descending*, which moves them further away. As shown in Table 13, descending order leads to noticeable performance degradation on multi-hop QA datasets such as 2WikiMQA and MuSiQue, indicating clear sensitivity to chunk placement and positional allocation. These results suggest that chunk reordering is a complementary factor to selective recomputation. Placing informative chunks closer to the query enables more effective information propagation during autoregressive decoding and is therefore adopted as the default setting in the proposed method.

| Model | Strategy | 2WikiMQA | MuSiQue | HotpotQA | NarrativeQA |
|---|---|---|---|---|---|
| *Qwen3-14B* | Baseline | 0.5161 | 0.3718 | 0.5922 | 0.1654 |
| | Ascending | 0.5058 | 0.3285 | 0.5972 | 0.2310 |
| | Descending | 0.4607 | 0.2875 | 0.6054 | 0.2295 |
| | $\Delta(\%)$ | -8.91% | -12.48% | +1.37% | -0.65% |

*Table 13.* Chunk reordering ablation on Qwen3-14B. Higher is better. $\Delta$ denotes the relative performance change from Ascending to Descending.

# F. More Implementation Details

**Norm Layer Selection.**   To determine which Transformer layer to use for identifying important tokens, we perform a layer-wise analysis on Qwen models. Specifically, we evaluate prompt–context attention norms extracted from different layers and measure their downstream impact on long-context retrieval accuracy. We find that attention norms from intermediate-to-late layers, particularly layers 22–25, consistently yield the strongest performance across context lengths. Based on this observation, we use layers 22–25 to compute prompt-conditioned attention norms and select recomputation targets for all models in our experiments. This choice reflects a practical trade-off between semantic maturity and positional sensitivity, and we observe that performance is relatively stable within this layer range.

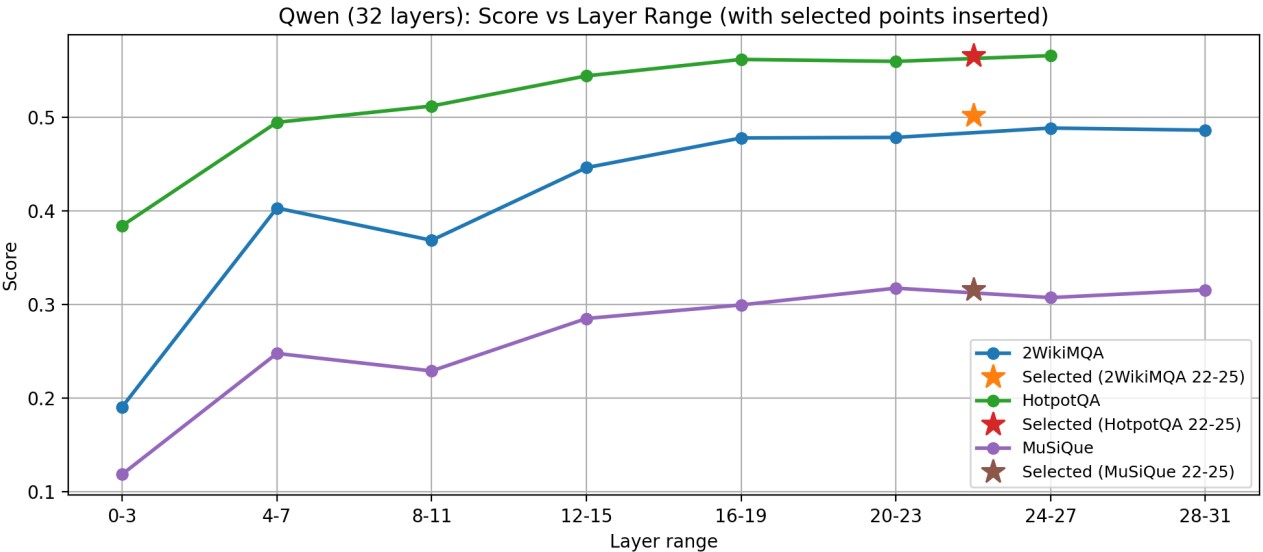

*Figure 4.* Downstream long-context QA score on Qwen3-14B (32 layers) as a function of the Transformer layer range used to compute prompt-conditioned attention norms for token selection, evaluated on 2WikiMQA, HotpotQA, and MuSiQue. Stars mark the layers 22–25 range adopted in our method. Higher is better.

**Recomputation Layers.**   Once recomputation targets are selected, we recompute the corresponding key–value states across all Transformer layers. This ensures that recomputed tokens are fully consistent with the global causal attention geometry throughout the model, rather than correcting positional mismatches at only a subset of layers. While partial-layer recomputation is possible, we find full-layer recomputation to be more stable and easier to integrate without introducing additional hyperparameters.

**Discrepancy between the single-GPU baseline and ring attention.**   In exact arithmetic, ring attention computes the same attention output as the single-GPU full-prefill baseline. In practice, however, we observe small numerical differences. The primary reason is that floating-point arithmetic is not associative: ring attention changes the order in which partial statistics (e.g., row-wise maxima and normalizers for softmax, and the subsequent weighted-sum reductions) are accumulated across sequence-parallel chunks, leading to slightly different rounding behavior compared to the single-GPU kernel. These deviations can be amplified under mixed precision, since inputs are typically stored in `bf16` and intermediate reductions may use different accumulation precision depending on the kernel implementation. Thus, the mismatch is a numerical artifact of reduction order and precision, rather than an algorithmic difference.

## G. Latency Breakdown

To further analyze the practical overhead of the proposed method, Table 14 reports a runtime breakdown of different pipeline components under identical hardware settings. The results show that most latency is still dominated by prefilling, while the additional scoring stage introduces only minor overhead. Moreover, recomputation accounts for a comparable portion of runtime to existing recomputation-based baselines under the same recomputation ratio.

Overall, these results suggest that the proposed method achieves improved information flow with moderate additional runtime cost, while maintaining favorable end-to-end TTFT efficiency.

| Component | Time (ms) | %TTFT |
|---|---|---|
| Prefill | 951 | 64% |
| Scoring | 22 | 2% |
| Recompute | 374 | 25% |
| Inference | 131 | 9% |

*Table 14.* Latency breakdown of the proposed method under sequence-parallel inference. Percentages are reported relative to total TTFT.

