# OpenReview forum: "InfoFlow KV: Information-Flow-Aware KV Recomputation for Long Context"
_ICML.cc/2026/Conference — ICML 2026 regular_

### Official Review · Reviewer_LeAo · 2026-03-03

**Soundness:** 3
**Presentation:** 3
**Significance:** 3
**Originality:** 2
**Overall Recommendation:** 4
**Confidence:** 4

**Summary:**

This paper studies selective KV recomputation for chunk-wise long-context inference in large language and vision–language models. It frames recomputation as an information-flow problem arising from the mismatch between chunk-local prefilling and global causal decoding. The authors propose selecting recomputation tokens using prompt-conditioned attention norms computed under a reconstructed global RoPE positional assignment, and optionally reordering retrieved chunks based on estimated importance. The method is evaluated on multiple long-context QA benchmarks for both LLMs and VLMs, with analyses of positional configurations and latency–accuracy trade-offs.

**Compliance With Llm Reviewing Policy:**

Affirmed.

**Final Justification:**

I thank the authors for the clear and constructive rebuttal. The additional explanations and experiments have addressed nearly all of my original concerns. In particular, the clarification of the paper’s scope makes the contribution more transparent and easier to evaluate.

Overall, the rebuttal meaningfully improved my confidence in the work. As a result, I am increasing my score from 2 to 4.

**Key Questions For Authors:**

- You can perform full-context prefilling in your experiments; what concrete deployment setting makes full prefilling infeasible and thus justifies the need for your method?
- Can you provide quantitative evidence that your approach more faithfully restores global causal interactions (e.g., approximation error to full-context attention) rather than only improving task accuracy?
- Can you clearly specify the chunk reordering algorithm, including how chunk-level importance is computed and how the final ordering is determined?

**Limitations:**

No. While the paper includes an Impact Statement, it focuses mainly on asserting that the method does not introduce additional societal risks beyond those of the underlying models. It does not meaningfully discuss the technical limitations of the approach (e.g., approximation errors, failure modes, or degradation under certain contexts) or potential negative impacts in deployment settings where approximate inference may reduce reliability. A more substantive discussion of methodological limitations and possible risks would improve the paper.

**Strengths And Weaknesses:**

# Strengths:
- The paper tackles a practically relevant problem in long-context inference, namely mitigating the mismatch introduced by chunk-wise prefilling when reusing KV caches under global causal decoding.
- The method is simple, model-agnostic, and inference-only, making it easy to integrate into existing LLM/VLM pipelines without retraining.

# Weaknesses:
- The main mechanism selecting tokens using attention-based importance scores and enforcing positional consistency under RoPE is not new. The contribution largely combines existing components (attention scoring, selective recomputation, positional alignment) in an incremental way, without introducing a substantially new algorithmic or modeling idea.
- Although the paper frames the contribution as mitigating “information-flow mismatch,” the method does not explicitly model information flow beyond using attention mass as a heuristic. There is no formal analysis of how causal interactions are restored or how closely the method approximates full-context inference, leaving the central claim only loosely supported.
- In Transformer-based long-context modeling, core challenges typically arise from inputs exceeding the trained context window or from hard resource constraints that make full attention infeasible. In this work, neither scenario is clearly established: the evaluated models can already perform full-context prefilling, and no concrete deployment setting is defined where full prefilling is impossible. As a result, the method mainly corrects approximation errors introduced by chunk-wise prefilling itself, rather than addressing a fundamental long-context limitation.
- While the stated motivation emphasizes mitigating mismatch, the evaluation is framed largely around speed–accuracy trade-offs rather than directly measuring mismatch reduction or approximation fidelity to full-context behavior. Performance gains over prior recomputation heuristics are generally modest and often remain below the full-context baseline. In addition, compute-aligned comparisons are not rigorously controlled, making it difficult to assess the practical significance of the improvements.
- The comparison set is narrow and focused mainly on closely related heuristic recomputation methods. Stronger or more diverse long-context approximation baselines are not included, and the related work discussion (especially Sections 2.2 and 2.3) is limited in scope. The paper does not sufficiently position the method within the broader literature on long-context inference, KV selection/eviction, sparse attention, or approximation-based decoding, making it unclear how the work advances beyond existing attention-based heuristics.
- The chunk reordering procedure (Section 4.3) is not clearly specified: the aggregation from token-level scores to chunk-level importance and the exact reordering mechanism are not formalized, which reduces clarity and reproducibility.

---

> ### Author Rebuttal · Authors · 2026-03-31
>
> 1. Novelty Concern
>
>     While our method uses components like attention-based scoring and selective recomputation, our contribution is not a simple combination of existing techniques, but addressing a **previously overlooked setting** and introducing a **new guiding principle**.
>
>     Specifically, prior work treats token selection as a relevance estimation problem under a fixed positional configuration. We show that, in **multi-chunk settings**, positional geometry for prompt and contexts under RoPE may affect downstream performance.
>
>     Our key contribution is a **RoPE-consistent selection principle**, which governs how tokens should be selected and positioned to remain compatible with inference-time geometry which is under explored in previous literature. This principle further enables operations such as reordering, and leads to **consistently improvements over other methods**.
>
> 2. Information Flow Clarification
>
>     In our work, “information flow” refers to whether selected tokens can effectively influence downstream generation through the attention mechanism. We do not attempt to explicitly model causal interactions. Instead, we adopt attention-based signals as a practically grounded proxy for identifying tokens that contribute to generation.
>
>     Our contribution is therefore not to provide a formal analysis of interaction recovery, but to show that **token selection must be consistent with the inference-time positional geometry (e.g., RoPE)** in order for these signals to remain reliable. We demonstrate that when this consistency is violated (e.g., in multi-chunk settings), selection quality degrades, while enforcing RoPE-consistent alignment significantly improves downstream performance.
>
>     From this perspective, our method is better understood as restoring a RoPE-consistent structure that enables effective generation under constrained computation.
>
> 3. Scope Concern
>
>     Our work does not target the classical scenario where full-context attention is impossible due to context length limits. Instead, we focus on a **practically important regime** where context is processed in **multiple independently prefetched chunks**, which commonly arises in systems such as retrieval-augmented generation (RAG) and agent pipelines. In these settings, documents are often preprocessed into KV caches, and **recomputing full-context prefilling is unnecessary or inefficient**. Our goal is therefore to facilitate effective generation under this widely adopted workflow, rather than replace full prefilling.
>
>    Second, even when full prefilling is feasible, our method provides practical efficiency advantages. As shown in Fig. 2 and Table 5, it achieves lower latency than full-context processing while maintaining competitive performance, demonstrating a favorable efficiency–performance tradeoff compared to other methods.
>
>     Finally, our formulation is aligned with prior work such as CacheBlend and EPIC, which operate under similar assumptions. We do not claim to introduce a new problem setting, but rather to **improve upon existing approaches within this regime**, where we consistently achieve stronger performance.
>
> 4. Performance Concern
>
>     Please refer to review x8kc section Performance Clarification for our detailed analysis and comparison with baseline.
>
> 5. More Baselines
>
>     We clarify that we have **different problem formulations**. KV eviction methods aim to reduce **KV memory**, while sparse attention and approximation methods accelerate inference by modifying or approximating attention. In contrast, our work focuses on **multi-context KV reuse**, where independent chunk must be selectively recomputed and reorganized to support downstream generation.
>
>     These approaches are therefore complementary in functionality. Our method operates at the prefill stage; after recomputation, the pipeline reduces to a standard decoding process, which can be directly combined with KV compression or sparse-attention techniques.
>
>     Among these, KV-based methods are the most closely related, as they also aim to manage or update KV representations. We have included representative baselines from this category, and our method consistently achieves **stronger performance under comparable settings**. All three methods degrade due to irreversible token loss, whereas our approach instead reconstructs KV representations to better support downstream generation.
>
>
> |Method|HotpotQA|2WikiMQA|MuSiQue|NarrativeQA|
> |---|---|---|---|---|
> |Baseline(fullKV)|0.5410|0.4588|0.3285|0.1862|
> |NoRecompute|0.4253|0.3066|0.2462|0.2810|
> |Ours|0.5123|0.4208|**0.2996**|0.3141|
> |Ours+Reorder|**0.5202**|**0.4417**|0.2793|**0.3243**|
> |PyramidKV|0.2619|0.2547|0.1457|0.2932|
> |SnapKV|0.2585|0.2643|0.1498|0.2911|
> |StreamingLLM|0.2257|0.2423|0.1114|0.1932|
>
> 6. Reordering Clarification
>
>   Please refer to the Chunking Reordering Specification section in our response to Reviewer nJcD for more details.

---

> > ### Author Rebuttal · Reviewer_LeAo · 2026-04-01
> >
> > Thank you for the detailed clarification of the intended deployment setting and for explaining the motivation behind the RoPE-consistent selection principle. The rebuttal helps better position the method within multi-chunk KV reuse workflows and clarifies how the approach differs from standard KV eviction or sparse-attention techniques.
> >
> > However, several core concerns remain only partially addressed. In particular, the rebuttal does not clearly establish when full-context prefilling becomes sufficiently inefficient or impractical to justify the proposed mechanism, given that full-context processing is feasible in the presented experiments. My concern about the reliance on full-context processing also stems from a practical scenario: when the context window is truly insufficient and full-context attention is no longer possible, it is unclear whether the proposed method can still function effectively, since the current formulation appears to depend on access to full-context computation during recomputation or alignment.
> >
> > In addition, while the method is framed as mitigating information-flow mismatch, the evaluation still focuses primarily on downstream task accuracy rather than directly measuring approximation fidelity to full-context behavior. As a result, it remains difficult to assess how broadly necessary and robust the proposed approach is across different long-context regimes.
> >
> > Overall, the rebuttal improves the conceptual framing of the work, but questions about the necessity, robustness, and applicability of the method remain partially unresolved.

---

> > > ### Author Response · Authors · 2026-04-04
> > >
> > > We thank the reviewer for the follow-up. We believe this concern may stem from a difference in the assumed setting of our work.
> > >
> > > 1. Scope Related
> > >
> > > Our work targets a **practically important and increasingly prevalent setting** in modern LLM systems, where context is **prefetched and stored as independent KV chunks** to improve serving efficiency. This workflow is widely adopted in applications such as retrieval-augmented generation (RAG) and agent pipelines, where a large portion of the context (e.g., retrieved documents or knowledge sources) is **shared across queries**, while the query itself is dynamic.
> > >
> > > However, because context chunks are processed independently, their interactions are **not directly captured** in the cached representations, leading to limited cross-chunk interaction during generation. As a result, **selectively recomputing a subset of tokens to partially update the reused KV caches** becomes a **natural and necessary design choice** for practical deployment. In contrast, full-context prefilling introduce **substantial redundant computation** to process the same context multiple times.
> > >
> > > This formulation is consistent with prior context-caching approaches such as CacheBlend and LegoLink (EPIC), which operate under the same regime.
> > >
> > > 2. Efficiency Analysis
> > >
> > > Under our formulation, full prefilling corresponds to recomputing 100% of tokens and serves as an **oracle upper bound**, rather than a competing solution within the same efficiency regime. We show below the recompute overhead for 15% on H200 and fully prefill where selective recomputation significantly reduces the recompute overhead across all sequence lengths. For example, at 131K tokens, recomputation latency is reduced from **14.7s to 2.7s**. This highlights that full prefilling introduces substantial redundant computation, especially at longer context lengths.
> > >
> > > |SeqLen|Ratio(%)|Recompute(ms)|
> > > |---|---|---|
> > > |32k|15|**333**|
> > > ||100|1616|
> > > |65k|15|**825**|
> > > ||100|4704|
> > > |131k|15|**2711**|
> > > ||100|14711|
> > >
> > > To evaluate in an end-to-end manner, we have conducted **extensive experiments** (Fig. 2, Table 5, and the *Efficiency Clarification* in response to reviewer nJcD). In these experiments, our approach achieves **lower latency on sequences of length up to 30K** while maintaining competitive performance.
> > >
> > > To further test efficiency on a longer contexts where full prefilling becomes increasingly inefficient, we adopt a **sequence-parallel (SP) inference framework** using 4× H200 to enable more GPU memory(as discussed in Section 7).  We use LongBench V2, with average context lengths of **~84K tokens** for the short and medium subsets and **over 130K tokens** for the long subset. These tasks cover a **diverse range of long-context scenarios**, including Single/Multi-Doc QA, document/data/code understanding, and long in-context learning, proving the applicability of our method. The results are summarized in Table below.
> > >
> > > Our method reduces latency to **~60% of ring attention**, achieving **2.0s savings** on the short and medium subsets and **4.7s savings** on the long subset, while maintaining comparable, or better accuracy. This demonstrates a strong efficiency–performance tradeoff.
> > >
> > > Notably, the efficiency gains become more pronounced as the context length increases, highlighting the advantage of our approach in long-context scenarios. Overall, these results demonstrate **substantial practical acceleration** over the baseline, translating directly into improved efficiency for real-world model serving.
> > > |Longbench|Method|Accuracy(%)|TTFT(ms)|
> > > |---|---|---|---|
> > > |Short+Medium|RingAttention(Baseline)|**34.52**|5347|
> > > ||Ours|33.74|**3274**|
> > > |Long|RingAttention(Baseline)|27.10|11373|
> > > ||Ours|**28.04**|**6589**|
> > >
> > > 3. Relation to Full-context Approximation.
> > >
> > > In our formulation, we do **not aim to approximate full-context attention**. Instead, our *information flow* refers to whether identified tokens, using attention-based signals under **inference-time consistent positional geometry,** can effectively influence downstream generation. Therefore, evaluating our method based on approximation fidelity to full-context attention is **not aligned with our objective**, as the two target fundamentally different properties.
> > >
> > > Nevertheless, to provide additional insight, we also evaluate how well different methods **recover attention patterns** compared to the full-context case. We report the average attention matrix mse across all layers of 15% recompute ratio using 100 samples from HotpotQA, our method achieves the **lowest attention MSE**(0.88e-7, 78.3% reduction).
> > >
> > > This result suggests that, although full-context approximation is not our design goal, our method is still able to **more effectively recover the underlying attention structure**, leading to improved information propagation and downstream performance.
> > >
> > > |Method|**AttentionMSE(1e-7)**|↓Reduction|
> > > |---|---|---|
> > > |NoRecompute|4.03|—|
> > > |Ours|0.88|**78.3%**|
> > > |CacheBlend|1.23|68.4%|
> > > |EPIC|2.05|49.1%|

---

### Official Review · Reviewer_X8kc · 2026-03-07

**Soundness:** 2
**Presentation:** 3
**Significance:** 2
**Originality:** 2
**Overall Recommendation:** 4
**Confidence:** 4

**Summary:**

This paper addresses the challenge of long-context inference in large language models (LLMs) under memory-constrained settings. The authors propose an information-flow-aware KV recomputation framework that operates in a chunk-wise fashion. The core idea is to first process a long input by splitting it into chunks and computing compressed KV caches (e.g., via token eviction), and then selectively recompute full KV caches for a subset of chunks deemed most relevant to the given query. To identify which chunks to recompute, the paper introduces a prompt-conditioned attention norm, which measures the information flow from context chunks to the prompt by aggregating attention weights and value norms across layers and heads. To address the positional inconsistency introduced by recomputing only selected chunks in isolation, the authors propose a global positional reconstruction strategy that reassigns RoPE-based positional indices to preserve the original global ordering of tokens. Additionally, a chunk reordering mechanism is introduced to place selected chunks closer to the query in the recomputed sequence, motivated by the observation that proximity to the query in positional space can benefit attention computation.

**Compliance With Llm Reviewing Policy:**

Affirmed.

**Final Justification:**

The authors have addressed all my concerns. I've raised my score.

**Key Questions For Authors:**

Please see the above comments.

**Limitations:**

Please see the above comments.

**Strengths And Weaknesses:**

Strengths:
1. The prompt-conditioned attention norm provides a principled and computationally lightweight way to estimate the relevance of each context chunk to the query.
2. The paper includes a relatively comprehensive set of ablations covering selection ratio, chunk granularity, reordering strategies, and the geometric properties of RoPE that motivate positional reconstruction.

Weaknesses:
1. The paper frames the prompt-conditioned attention norm as capturing "information flow," but the metric is essentially a weighted combination of attention scores and value norms computed during the first-pass chunked inference. True information flow analysis in transformers would involve tracking how representations propagate and transform across layers (e.g., through residual streams), not merely aggregating per-layer attention statistics. The terminology elevates what is a reasonable but straightforward heuristic into something that sounds more theoretically grounded than it actually is.
2. The prompt-conditioned attention norm is computed from the first-pass inference, which itself uses aggressively compressed (evicted) KV caches. If the first pass already loses critical tokens due to eviction, the resulting attention patterns may be distorted, meaning the chunk selection metric is built on a degraded foundation. The paper does not analyze or discuss how the quality of the first-pass eviction policy affects the reliability of the selection metric. A failure-case analysis showing when chunk selection goes wrong would strengthen the paper.
3. The paper positions itself within the KV cache management literature but does not compare against retrieval-augmented generation (RAG) pipelines or embedding-based retrieval methods that also address long-context QA by selecting relevant passages. Since the chunk selection step is functionally similar to a retrieval step, comparing with BM25 or dense retriever baselines for the selection component would help contextualize the contribution.
4. The paper claims the method is efficient, but the discussion of computational overhead is largely qualitative. There is no wall-clock time comparison between InfoFlow and baselines under identical hardware conditions. Recomputing full KV caches for selected chunks requires a second forward pass through potentially large portions of the context. The paper does not clearly quantify what fraction of the original context is typically recomputed, nor does it provide a latency breakdown (first-pass time vs. selection time vs. recomputation time). Without this, the practical efficiency gains remain unclear.
5. The paper proposes moving selected chunks closer to the query in positional space, arguing that RoPE's distance decay means closer tokens receive higher attention. However, this reordering fundamentally alters the sequential semantics of the input. For tasks requiring temporal or logical reasoning over the ordering of events in the context, this reordering could be harmful. The paper does not evaluate on tasks where document order is critical, and the ablation showing reordering helps on average does not rule out significant degradation on order-sensitive subsets.
6. Looking at the detailed benchmark results, the improvements are not uniform across tasks. On some subtasks (e.g., certain categories in LongBench or RULER), the gains are marginal or the method occasionally underperforms baselines. The paper tends to emphasize average scores and does not provide a thorough discussion of when and why the method fails. A per-category breakdown with error analysis would be more informative than aggregate numbers alone.

---

> ### Author Rebuttal · Authors · 2026-03-31
>
> 1. More on Information Flow
>
>    We refer to our response to Reviewer LeAo for a clarification of our operational definition of *information flow*, which does not aim to model formal causal interactions or representation propagation across layers. Instead, we use attention-based signals as a practical proxy for identifying tokens that influence downstream generation.
>
>     To further support that our method captures **layer-consistent importance**, we analyze its cross-layer agreement. We construct a consensus set of important tokens (selected by at least half of all layers), and measure how well tokens selected from a small subset of layers (e.g., layers 22–25) cover this consensus.
>
>     As shown in Fig(https://anonymous.4open.science/r/icml-rebuttal-E210/), the resulting consensus recall is consistently high with low variance across samples, indicating that the selected tokens capture signals shared across layers. This suggests that our method naturally aligns with a layer-wise notion of importance, that are effective for guiding token selection.
>
> 2. First-Pass Eviction Clarification
>
>     We would like to clarify that our method does **not** perform an initial inference pass with any eviction. Instead, the selection metric is computed from the **prefetched chunks**. After this selection step do we perform recomputation for the selected tokens. Therefore, we do not apply any eviction or compression throughout our pipeline.
>
> 3.  RAG Clarification
>
>     Our work focuses on post-retrieval KV cache management rather than retrieval itself. Datasets like HotpotQA and 2WikiMQA already provide context, and we optimize token-level selection within it. Thus, retrieval-based methods are therefore largely orthogonal to our problem.
>
>     That said, we additionally consider BM25 Recompute, which replaces token-level selection with passage-level selection under the same budget. BM25 Recompute consistently underperform our method across datasets, indicating that chunk-level retrieval cannot substitute for token-level selection. These results show that our method cannot be replaced by naive retrieval.
>
> | Method        | HotpotQA   | 2WikiMQA   | MuSiQue    | NarrativeQA |
> | ------------- | ---------- | ---------- | ---------- | ----------- |
> | Qwen3-14b     | 0.5922     | 0.5161     | 0.3718     | 0.1654      |
> | Our           | 0.5954     | 0.5019     | **0.3386** | 0.2288      |
> | Our+Reorder   | **0.5972** | **0.5058** | 0.3285     | **0.2310**  |
> | BM25Recompute | 0.4153     | 0.1904     | 0.1117     | 0.1874      |
>
> 4. Run-time Breakdown
>
>     We clarify that our experiments already report **wall-clock latency (TTFT) under identical hardware conditions** (Fig. 2), where our method lies on the Pareto frontier compared to baselines.
>
>     To further quantify the practical overhead, we provide a runtime breakdown (table below). The results show that components introduced by our method (scoring + recomputation) account for only ~30% of the total latency. Moreover, the recomputation time is consistent across baselines with same recomputation ratio, leaving only 2% overhead for our method.
>
>     These results indicate that the additional cost introduced by our method is moderate and does not negate its efficiency benefits. We will clarify these points and include more explicit discussion of runtime composition in the revision.
>
> |Component|Time|%ofTTFT|
> |---|---|---|
> |Prefill|951ms|69%|
> |Scoring|22ms|2%|
> |Recompute|374ms|27%|
> |Inference|131ms|10%|
>
> 5. Reordering Constraints
>
>     We agree that reordering can be harmful in tasks where the sequential structure of the input is critical. Importantly, as described in Section 4.3, reordering is an optional component, intended only for settings where input chunks are independent. When ordering carries semantic meaning, reordering is not applied. We also include a fixed-chunk setting in Table 3, where the independence assumption is weakened. In this case, reordering does not outperform the non-reordered variant, consistent with the reviewer’s concern, while still outperforming other baselines.
>
>     Therefore, reordering should be viewed as a conditional enhancement, whose effectiveness depends on the structural properties of the input.
>
> 6. Performance Clarification
>
>     We would like to clarify that the “baseline” in our comparison corresponds to **full KV prefilling without any recomputation**, which is effectively an **oracle upper bound** (i.e., equivalent to recomputing 100% of tokens). As such, it serves as  **maximum achievable performance without any efficiency constraint** and gaps to this baseline are expected under constrained budgets
>
>     More importantly, when compared against **existing recomputation methods**, our approach consistently achieves **stronger and more robust performance across benchmarks**, often significantly outperforming prior methods.

---

> > ### Author Rebuttal · Reviewer_X8kc · 2026-04-01
> >
> > The rebuttal clarifies my concerns. I will raise my score accordingly.

---

### Official Review · Reviewer_L3tp · 2026-03-08

**Soundness:** 4
**Presentation:** 3
**Significance:** 3
**Originality:** 2
**Overall Recommendation:** 5
**Confidence:** 3

**Summary:**

Key-value (KV) cache prefilling dominates the inference efficiency for long-context retrieval-augmented generation. Existing methods accelerate the inference by first precomputing KV caches for individual documents offline and then selectively recomputing KV cache for a small subset of tokens to capture cross-document interactions. However, these methods typically fail to fully capture (i) semantic relevance, particularly with respect to the query; (ii) effective information flow, i.e., whether a token is structurally positioned to influence downstream decoding under the global causal attention graph. This paper proposes to address the insufficiencies of previous methods by (i) selective KV re-computation with an attention-norm criterion from the query to context tokens (ii) a retrieved-chunk reordering strategy based on identified information-flow–critical tokens. Extensive experiments on LLM and VLM benchmarks demonstrate the superior performance of the proposed approach.

**Compliance With Llm Reviewing Policy:**

Affirmed.

**Final Justification:**

The rebuttal addressed my main concerns and better demonstrated the advantages of the proposed approach with more ablation studies and baselines.

**Key Questions For Authors:**

I'm a bit confused with Table 4. Does $k$ here correspond to the top-k selection in section 4.2? Why do we generally see worse results with a larger $k$?

**Limitations:**

The proposed approach may be limited due to:
- The need for re-computation across queries compared to query-agnostic approaches
- Inherent logical or semantic connections across chunks that are not directly clear from the query
- Abstract / non-factual queries like summarization / deep research settings

**Strengths And Weaknesses:**

# Strengths

**S1.** The paper is overall easy to follow.

**S2.** The proposed approach is simple, general and principled.

**S3.** Empirical studies are extensive and cover both LLMs and VLMs. They cover the study of accuracy-TTFT trade-offs and the scaling behavior as the context size increases.

# Weaknesses

**W1.** While the KV reorder strategy is clear in Figure 1, the exposition for it in Section 4.3 is insufficient and not clear enough.

**W2.** The proposed approach is constrained to RoPE.

**W3.** The paper still misses several highly relevant related work on KV cache strategies for RAG, positional encoding configuration, parallel encoding, etc. E.g.,

- KVLink: Accelerating Large Language Models via Efficient KV Cache Reuse
- APE: Faster and Longer Context-Augmented Generation via Adaptive Parallel Encoding
- Block-Attention for Efficient Prefilling
- TurboRAG: Accelerating Retrieval-Augmented Generation with Precomputed KV Caches for Chunked Text
- Eliminating Position Bias of Language Models: A Mechanistic Approach
- Parallel context windows for large language models

**W4.** The paper does not perform ablation studies for KV / chunk reorder, e.g., retriever-ranked descending ordering

---

> ### Author Rebuttal · Authors · 2026-03-31
>
> 1. Chunk Reordering Specification
>
>     We thank the reviewer for raising this up, please refer to the *Chunking reordering Specification section* in our formal response to Reviewer nJcD, and we will further specify this in our next revision.
>
> 2. Restriction to RoPE-based Architectures
>
>     **First, while our analysis is instantiated under RoPE, the underlying methodology is not specific to RoPE itself.** Our framework studies KV selection in multi-chunk settings by analyzing how positional encoding geometry—across prompt and context segments—shapes token interactions and downstream performance. We show that maintaining consistency with the inference-time geometry is critical for effective information flow. This perspective can be extended to other positional encoding schemes by considering their induced interaction patterns and not tied to RoPE.
>
>     **Second, we focus on RoPE because it is the dominant positional encoding used in modern LLMs and VLMs.** Most widely used open-source models, including LLaMA, Qwen, and GLM families, adopt RoPE or its variants. To the best of our knowledge, this covers the vast majority of current large-scale deployments.
>
>     Therefore, we believe our analysis is general and our experiment on RoPE is representative.
>
> 3. More Relevant Work
>
>     We appreciate the suggestion with more baselines to further showcase of the advantage of our method. Below we include three more methods of KVLink, TurboRAG, and PWC at the tables below(with ***1st***  and **2nd** highlighted)
>
> KVLink: We comapre KVlink5 with our methods using Llama3-3b model on four different benchmarks. Result have shown that our method achieve comparable performance with KVLink. However, KVLink5 is reported to require extra training of over 3M datapoints and potential sensitivity to data and domain shifts. In contrast, our approach is fully **training-free,** making it directly applicable across models and tasks without retraining.
>
> |Method|HotpotQA|2WikiMQA|MuSiQue|NarrativeQA|Avg|
> |---|---|---|---|---|---|
> |Llama3-3b|0.5131|0.4073|0.3293|0.2762|0.3815|
> |NoRecompute|0.3863|0.2218|0.2161|0.2391|0.2658|
> |Ours|**0.4652**|**0.4007**|***0.2693***|0.2574|**0.3482**|
> |Ours+Reorder|***0.4994***|0.3661|0.2522|**0.2742**|**0.3480**|
> |KVLink5|0.4551|***0.4049***|**0.2555**|***0.2766***|**0.3480**|
>
>
> TurboRAG and PWC: We have tested these two training-free approach on two different models -Qwen3-14b and Llama3.1-8B-Instruct. Across both models, our method achieves consistently strong performance relative to prior KV selection and recomputation baselines, supporting the effectiveness of our RoPE-consistent formulation in enabling effective information flow during generation. Overall, the results demonstrate that our method delivers **consistent improvements across models and datasets**, supporting the effectiveness of our design.
>
> |Method|HotpotQA|2WikiMQA|MuSiQue|NarrativeQA|Avg|
> |---|---|---|---|---|---|
> |Llama-3.1-8B-Instruct|0.5410|0.4588|0.3285|0.1862|0.3786|
> |NoRecompute|0.4253|0.3066|0.2462|0.2810|0.3148|
> |Ours|**0.5123**|**0.4208**|***0.2996***|0.3141|**0.3867**|
> |Ours+Reorder|***0.5202***|***0.4417***|0.2793|**0.3243**|***0.3914***|
> |TurboRAG(1024)|0.4497|0.3640|0.2600|0.2190|0.3232|
> |PCW|0.4145|0.3073|0.2388|***0.3292***|0.3224|
>
> |Method|HotpotQA|2WikiMQA|MuSiQue|NarrativeQA|Avg|
> |---|---|---|---|---|---|
> |Qwen3-14b|0.5922|0.5161|0.3718|0.1654|0.4114|
> |NoRecompute|0.3059|0.1162|0.1012|0.2078|0.1828|
> |Ours|**0.5954**|**0.5019**|***0.3386***|0.2288|***0.4162***|
> |Ours+Reorder|***0.5972***|***0.5058***|**0.3285**|**0.2310**|**0.4156**|
> |TurboRAG(1024)|0.3337|0.2880|0.0723|0.0490|0.1858|
> |PCW|0.3806|0.1939|0.1541|***0.2727***|0.2503|
>
> 4. KV Reordering Ablation
>
>     We compare two chunk reordering strategies on Qwen3-14B: **Ascending**, which places high-importance chunks closer to the query, and **Descending**, which moves them further away. On multi-hop QA datasets (2WikiMQA, MuSiQue), Descending leads to a **notable relative drop in F1 (up to 12.5%)**, indicating clear sensitivity to chunk placement. This suggests that chunk reordering is a critical complementary factor. Ascending order contribute more effectively during generation and we take it as our default setting.
>
> | Dataset | Baseline | Ascending | Descending | Δ(%) |
> | --- | --- | --- | --- | --- |
> | 2wikimqa | 0.5161 | **0.5058** | 0.4607 | -8.91% |
> | hotpotqa | 0.5922 | 0.5972 | **0.6054** | +1.37%   |
> | musique | 0.3718 | **0.3285** | 0.2875 | -12.48%  |
> | narrativeqa | 0.1654 | **0.2310** | 0.2295 | -0.65% |
>
> 5. Specification on **k** in Tab.4
>
>     Here, k refers to the number of patch groups the original image is divided into, rather than the top-k token selection in Section 4.2. A larger k corresponds to more and smaller segments of the image, which can break spatial coherence and disrupt long-range dependencies, leading to degraded performance.

---

> > ### Author Rebuttal · Reviewer_L3tp · 2026-04-02
> >
> > Thank you for the detailed responses and additional experiment results. I've raised my score accordingly, and I don't have further questions or concerns. I think it will be helpful to integrate some of the explanations into the manuscript to further improve the clarity of the work.

---

### Official Review · Reviewer_nJcD · 2026-03-09

**Soundness:** 3
**Presentation:** 3
**Significance:** 3
**Originality:** 2
**Overall Recommendation:** 4
**Confidence:** 4

**Summary:**

This paper studies selective KV recomputation for long-context inference under chunk-wise prefilling. The core idea is to score context tokens using prompt-conditioned attention, recompute only a small subset of important tokens under the global context, and use global positional reconstruction so that token selection is more consistent with inference-time RoPE geometry. The paper also proposes an optional chunk reordering strategy and evaluates the method on both LLM and VLM benchmarks.

**Compliance With Llm Reviewing Policy:**

Affirmed.

**Key Questions For Authors:**

1，The method seems fairly close in spirit to FINCH at the level of chunked processing and prompt-guided token scoring. Could the authors clarify this connection more explicitly and better isolate the main novelty of the paper?

2.Is the proposed recomputation procedure compatible with FlashAttention-style optimized kernels? If not, how should the reported speedups be interpreted relative to a dense baseline that can fully benefit from such kernels?

3.For the chunk reordering module, how exactly is the chunk-level importance score computed, and is the same token budget used for each chunk during the first-stage selection?

4.Could the authors add results on a more recent benchmark such as LongBench v2?

**Limitations:**

yes

**Strengths And Weaknesses:**

This paper addresses an important problem in long-context inference, where prefilling often dominates the actual decoding cost. The proposed method is simple, training-free, and reasonably well motivated. I also appreciate that the paper includes ablations on RoPE geometry and some efficiency analysis, rather than only reporting end-task numbers.

Empirically, the method appears consistently competitive against prior selective recomputation baselines, and the paper does a good job highlighting why positional consistency matters for token selection. The global positional reconstruction and the discussion around RoPE geometry are useful parts of the paper.

Weaknesses:

* The originality is somewhat limited. The overall idea appears related to prior chunked long-context processing and prompt-guided token/KV selection methods, especially FINCH. I think the paper would benefit from a more careful discussion of what is genuinely new here.

* The efficiency claim needs clearer qualification. It is not clear whether the proposed recomputation procedure is compatible with highly optimized kernels such as FlashAttention. If it is not, then the practical speed advantage may be weaker than suggested, since a dense baseline that fully benefits from such kernels could be faster in practice.

* The chunk reordering component is not fully specified. In particular, the paper does not clearly formalize how the chunk-level importance score is computed from the first-stage token selection, or whether the same token budget is used for every chunk.

* The evaluation could be strengthened with a more recent benchmark, for example LongBench v2, to show whether the method remains effective on newer and potentially more challenging long-context settings.

---

> ### Author Rebuttal · Authors · 2026-03-31
>
> 1. Comparison with prior prompt-guided methods like FINCH
>
>     We thank the reviewer for pointing out the connection to prior KV selection methods such as FINCH. While these approaches share a high-level similarity in selecting tokens, our work differs fundamentally in both **perspective and mechanism**.
>
>     First, we formulate KV selection from an information flow perspective, focusing on whether tokens can effectively influence downstream generation rather than relying on correlation-based signals. This requires explicitly accounting for the RoPE-induced attention geometry, which is largely ignored in prior heuristic methods.
>
>     Second, we address a critical setting where context is composed of multiple independent chunks. In this scenario, positional inconsistency under RoPE degrades token interactions. We show (Sec. 4.2) that aligning selection with inference-time RoPE geometry is essential, leading to a global consistency principle. This principle governs both token selection and chunk reordering, which are not acknowledged by prior KV compression/eviction methods.
>
>     Overall, our work shows that effective KV selection must be **RoPE-consistent to preserve information flow**, providing a principled framework beyond relevance-based heuristics
>
> 2. Efficiency Clarification
>
>     Our method is not implemented with FlashAttention, as selective recomputation introduces irregular attention masks that are not well supported. However, this does not affect the validity of efficiency. In our experiments, the full-prefill baseline uses FlashAttention, while our method uses FlashInfer, a widely adopted solution for sparse or irregular attention. Therefore, the TTFT results reflect a comparison between **optimized implementations on both sides.**
>
>     To further strengthen this point, we additionally compare on multi-gpu inference case against an optimized **4-GPU** **ring attention** baseline. The results show that our speed advantage persists and becomes larger as context length increases starting mildly(15ms) at 8K and end up substantial(1400ms) at 32k. This trend supports our efficiency claim: as context grows, avoiding full KV communication and recomputing only a small subset of information-critical tokens becomes increasingly beneficial.
>
>     We agree that this qualification should be stated more clearly in the paper, and we will revise the text in our next revision.
>
>     | **Seq Len** | **Method** | **Recompute Ratio** | **TTFT (ms)** |
>     | --- | --- | --- | --- |
>     | 8192 | Single-GPU Prefill | – | 566.7 |
>     |  | Ring Attention | – | 247.5 |
>     |  | Ours | 0.15 | **232.0** |
>     | 16384 | Single-GPU Prefill | – | 1285.8 |
>     |  | Ring Attention | – | 707.8 |
>     |  | Ours | 0.15 | **427.6** |
>     | 32768 | Single-GPU Prefill | – | 3190.5 |
>     |  | Ring Attention | – | 2350.1 |
>     |  | Ours | 0.15 | **914.0** |
> 3. Chunk reordering Specification
>
>     We clarify our token selection and reordering pipeline below to avoid potential misunderstanding.
>
>     (1) We first score tokens **globally but under local RoPE**, where each chunk is prefetched independently with positions encoded $p_i^{\text{local}} \in \{0, \dots, |c_i|-1\}$. We then select the top 15% tokens across all chunks, a **single global budget**, without any per-chunk quota.
>
>     (2) We reorder chunks by the ratio of selected tokens to total chunk length; chunks with higher ratios, i.e., more informative chunks, are placed closer to the query.
>
>     (3) We realign positions under GLOBAL RoPE by assigning tokens to a shared range $\tilde p_j \in \{0, \dots, N-1\},$ , where N is the whole context length, following the final concatenated order. Then we reselect important tokens under the updated positional geometry, also a global budget.
>
>     (4) Finally, we recompute only the selected tokens.
>
>     The key principle is enforcing **RoPE-consistent geometry between selection and inference**.
>
> 4. More Benchmark
>
>     To further test robustness, we extended our evaluation to **LongBench v2 using Qwen3-14B**. We report results on the **180 samples of short-question subset** of the benchmark(10k~210k) which can fit into a single GPU. We made this choice because many of the longer examples require substantial truncation under a single-GPU full-prefill baseline, and in that regime the baseline performance is often close to random guessing, making the comparison less informative.
>
>     The results show that our method remains competitive. In detail, our method remains substantially above other alternative methods with a accuracy of over 28.89%, showcasing our advantage is general to broad tasks and benchmarks. We will clarify this evaluation setup in next revision.
>
>     | Strategy | Accuracy |
>     | --- |---|
>     | Baseline | 38.33% |
>     | Our | **28.89%** |
>     | CacheBlend | 26.11% |
>     | No Recompute | 21.11% |
>     | Lego | 20.56% |

---

> > ### Author Rebuttal · Reviewer_nJcD · 2026-04-08
> >
> > My concerns have been adequately addressed. I will keep the score.

---

### Decision · Program_Chairs · 2026-04-30

**Decision:**

Accept (regular)

**Comment:**

This paper addresses an important problem in long-context inference by casting the selective KV recomputation as an information flow problem. Empirical results are comprehensive and promising. There have been abundant discussions during the rebuttal, and the authors' responses have adequately addressed all the concerns. After the rebuttal, the paper received consistently positive evaluations from 4 reviewers.